# Clinical utility of tumour mutational burden on efficacy of immune checkpoint inhibitors in malignant solid tumours: protocol for a systematic review and meta-analysis

Xuemei Xiang ,[1] Yunming Li,[2,3] Xiaoguang Yang,[2] Wang Guo,[2,3] Pengfei Zhou[2,4]

¹Basic Medical Laboratory, People's Liberation Army The General Hospital of Western Theater Command, Chengdu, Sichuan, China
²Department of Information, People's Liberation Army The General Hospital of Western Theater Command, Chengdu, Sichuan, China
³Department of Statistics, Southwest Jiaotong University, Chengdu, Sichuan, China
⁴School of Public Health, Southwest Medical University, Luzhou, Sichuan, China

**Correspondence to**
Dr Yunming Li;
lee3082@sina.com

## ABSTRACT

**Introduction** A major development in solid malignancy treatment is the application of immune checkpoint inhibitors (ICIs), which have produced durable responses and increased survival rates. However, the therapeutic effect of ICIs has great heterogeneity in patients with cancer. We propose a systematic review to evaluate the predictive value of tumour mutation burden (TMB) on efficacy of ICIs.

**Methods and analysis** A systematic literature search will be conducted in the PubMed, OVID, Web of Science, Embase and Cochrane Central Register of Controlled Trials Library databases up to 31 May 2022. We will compare the efficacy of ICIs between TMB high group and TMB low group in terms of the HRs of overall survival (OS) and progression-free survival (PFS), and the OR of the objective response rate/overall response rate (ORR). The HRs of PFS and OS, and the OR of ORR, will be measured by an inverse variance weighted fixed effects model ($I^2 \leq 50\%$) or a DerSimonian-Laird random effects model ($I^2 > 50\%$). In addition, subgroup analysis, sensitivity analysis, heterogeneity analysis and publication bias will be conducted. We plan to conduct a subgroup analysis on age, sex, area, number of patients (high/low TMB), cancer type, tumour size, stage, line of therapy, TMB sequencing method, type of immunotherapy and follow-up period.

**Ethics and dissemination** Ethical approval and informed consent are not needed, as the study will be a literature review and will not involve direct contact with patients or alterations to patient care. This systematic review is anticipated to be finished in December 2023, and the results will be published in a peer-reviewed journal.

**PROSPERO registration number** CRD42021262480.

## STRENGTHS AND LIMITATIONS OF THIS STUDY

⇒ This will be an update comprehensive systematic review focused on the tumour mutation burden (TMB) and the efficacy of immune checkpoint inhibitors (ICIs) for the prognosis of patients with solid tumours.
⇒ We plan to conduct a comprehensive subgroup analysis of the association between TMB and the efficacy of ICIs, including age, sex, area, number of patients (high/low TMB), tumour size, stage, TMB sequencing method, type of immunotherapy and follow-up period.
⇒ We will focus on the long-term efficacy of ICIs in patients with solid tumours.
⇒ We will search databases for studies published in English, while other languages may be ignored.

## INTRODUCTION

Immune checkpoint inhibitors (ICIs) have been shown to prolong response and increase survival rates in various solid tumours and haematologic malignancies. However, the efficacy of ICIs seems satisfactory in some patients and unsatisfactory in others,[1–6] suggesting the need to identify biomarkers that indicate which subgroups are candidates for malignancy immunotherapy. Nowadays, researchers have identified several potential biomarkers, such as, tumour infiltrating lymphocytes and programmed cell death ligand 1 (PD-L1, transcriptomic epigenetic signatures and oncogenic driver mutations.[7] Among them, tumour mutation burden (TMB) is likely to be a potential biomarker. TMB is broadly defined as the number of somatic mutations per megabase of interrogated genomic sequence.[8] TMB is a continuous variable and variability of TMB (ranging from 0.001/Mb to more than 1000/Mb) has been observed across and within cancer types.[9 10] It was suggested that a higher TMB increases the likelihood of generating immunogenic tumour neoantigens recognised by the host immune system.[11–13]

Retrospective evidence suggests that TMB can predict the efficacy of ICIs, and recent US Food and Drug Administration approval of pembrolizumab for the TMB—high tumour subgroup. However, the predictive value seems inconsistent in patients with different

tumour types, which may be associated with the degree of variability in TMB. Current investigations indicate that some cancer types have less variability in TMB such as lung and head and neck cancers, and some have greater variability such as colon, bladder and uterine cancers.[14] Studies are attempting to validate the long-term oncologic impact of TMB. Although numerous studies have revealed the exciting forecasting capability of TMB on the efficacy of the ICIs, negative results have also been reported, especially in long-term survival.[15–17] As far as we know, three meta-analyses reported the predictive value of TMB.[18–20] The sample size of the first two studies was small and the subgroup analysis was incomplete.[17 18] The latest meta-analysis published in 2019 including 29 studies, with a total of 4431 patients.[19] However, there is also a lack of evidence regarding the long-term efficacy of all types of tumours due to the insufficient number of studies and patients. It is not sufficient to seek out the best threshold for TMB, and there is no consensus regarding the use of this biomarker for in small-cell lung cancer. Moreover, in most studies, PD-(L)1 monotherapy were performed, and the research on combined therapy is also insufficient.

Hence, we propose an update to the evidence by conducting a comprehensive systematic review and meta-analysis to evaluate the value of TMB on the efficacy of ICIs in malignant solid tumours. We will also proceed overall subgroup analyses to determine the promising effects of ICIs.

## METHOD
### Materials and methods
We submitted this study protocol to PROSPERO (CRD42021262480). This systematic review and meta-analysis will be conducted in accordance with the Preferred Reporting Items for Systematic Review and Meta-Analysis Protocols.[21 22]

### Inclusion and exclusion criteria
We will include all prospective or retrospective studies that meet the following criteria.

### Population
We will include cohort or clinical trials assessing ICIs, such as PD-1/PD-L1, cytotoxic T lymphocyte-associated antigen-4 (CTLA-4), or their combination, or with chemotherapy, in patients with malignant solid tumours. A cut-off of ≥10 mutations per megabase (mut/Mb) was chosen to define the 'high TMB' patient population.

### Intervention
ICI treatment in patients with cancer with malignant solid tumours.

### Comparator
We will evaluate the efficacy of ICI therapy in the TMB high group and the TMB low group. The HRs of progression-free survival (PFS), the HRs of overall survival (OS), the OR of overall response rate (ORR) and their 95% are

reported in our article. Besides, we will calculate them using the sufficient data collected in studies.

### Outcome
► Association between different levels of TMB and response rate of ICIs in all kinds of malignant solid tumour types, including OS, PFS, DFS, RFS, DSS and others.
► Association of subgroup analysis between different levels of TMB and efficacy of ICIs, including age, sex, area, number of patients (high/low TMB), tumour size, stage, TMB sequencing method, type of immunotherapy or follow-up period.
► Correlations between TMB and clinicopathological features, such as tumour size, stage and metastasis.
The exclusion criteria will be as follows:
1. Review, comments, case reports, non-human study.
2. There is no control groups and analysis.
3. The data are incomplete.

### Search strategy
The PubMed, Ovid, Web of Science, Embase and the Cochrane Central Register of Controlled Trials databases will be searched from inception to 31 May 2022, using the MeSH terms 'Immune Checkpoint Inhibitors' and the related keywords 'Immune Checkpoint Inhibition', 'Immune Checkpoint Blockers', 'Immune Checkpoint Blockade', 'PD-L1', 'CTLA-4', 'PD-1' or the name of the drugs (ie, atezolizumab, pembrolizumab, nivolumab, durvalumab, ipilimumab, avelumab, tremelimumab), 'Mutational Burden' or 'Mutation Burden'. The languages will not be limited in our search strategy. The search strategy for Ovid is presented in table 1 and the full search strategies and the results of five databases are presented in online supplemental file.

### Data abstraction
XX and WG will independently assess the eligibility of reports from the title and/or abstract. A third reviewer, YL, will be consulted in case of inconsistent. We will select studies that meet the inclusion criteria for further analysis. For the included studies that have no insufficient data, we will ask for the original data from corresponding authors analysis. The following items will be extracted from all included studies: first author, study design, year of publication, median age, sex, TMB sequencing method, follow-up period, type of cancer, tumour size, stage, type of immunotherapy, TMB cut-off, number of patients (high/low TMB), area of patients and outcomes (PFS, ORR, OS, etc).

### Assessment of risk of bias in included studies and study quality
Two systematic review authors (YL and WG) will independently assess the risk of bias for each study using the Newcastle-Ottawa Scale (NOS). The NOS will be adopted to assess the quality of the included studies.[23] The total score ranges from 0 to 9, where 8–9 points indicates high

**Table 1** Search strategy (OVID)

| Item | Search strategy |
|------|-----------------|
| #1 | exp Immune Checkpoint Inhibitors/ |
| #2 | ((immunotherap*) or (immune checkpoint inhibit*) or (ICI) or (immune checkpoint inhibit*) or (ICIs) or (immune checkpoint block*) or (ICB) or (ICBs) or (pembrolizumab) or (avelumab) or (nivolumab) or (durvalumab) or (tremelimumab) or (atezolizumab) or (Ipilimumab) or (Cemiplimab) or (tiragolumab) or (Dostarlimab) or (Camrelizumab) or (PD-1) or (programmed death 1) or (PD-L1) or (programmed death-ligand 1) or (anti-PD-1) or (anti-PD-L1) or (CTLA-4) or (Cytotoxic T-lymphocyte antigen 4)).tw. |
| #3 | #1 OR #2 |
| #4 | ((Carcinoma) or (Neoplasms) or (Cancer) or (Tumour) or (Tumor)).tw. |
| #5 | #3 and #4 |
| #6 | ((mutation burden) or (mutational burden) or (mutation load) or (mutational load) or (TMB) or (TML)).tw. |
| #7 | #5 and #6 |

quality of a study, 5–7 points indicated medium quality and less than 5 points indicates poor quality.

## Assessment of publication bias

If at least 10 studies are included, we plan to use Egger's test and the funnel plot to estimate the potential publication bias by R V.4.0.2. $P<0.05$ will be considered to indicate significant publication bias.

## Assessment of heterogeneity

The $\chi^2$ test will be used to estimate heterogeneity in pooling analysis. Heterogeneity is considered to be statistically significant when $p<0.10$ in all qualitative tests. The $I^2$ test will be used to examine the proportion of total variation, with values of 25%, 50% and 75% indicating low, moderate and high heterogeneity, respectively. We plan to conduct a meta-regression to confirm the source of heterogeneity within R V.4.0.2. We also plan to conduct a subgroup analysis on age, sex, area, number of patients (high/low TMB), tumour size, stage, TMB sequencing method, type of immunotherapy or follow-up period.

## Sensitivity analysis

To determine the robustness of the pooled results, sensitivity analysis will be performed by examining individual studies using R V.4.0.2.

## Data synthesis

The primary outcome of our article is the comparison of the efficacy of ICIs between the TMB high group and TMB low group, which will be assessed by the HRs of PFS and OS, and the OR of ORR. Heterogeneity among individual studies will be evaluated by the Q test; $I^2>50\%$ and/or $p<0.10$ will be considered to indicate significant heterogeneity.[24] DerSimonian-Laird random effects model will be used to calculate the pooled ORs or HRs with z test when significant heterogeneity is identified. Otherwise, inverse variance weighted fixed effects model will be adopted. In addition, to evaluate publication bias, the Begg's test and Egger's test will be applied. Funnel plots will be constructed. In addition, the stability of the results in our article will be tested by sensitivity analysis.

To further explore the variation in the effect of TMB on immunotherapy efficacy, subgroup analyses stratified by follow-up period, tumour size, tumour area, stage, line of therapy, TMB sequencing method, type of immunotherapy of ICIs alone (PD-L1, PD-1, CTLA-4, etc) or ICIs combined with chemotherapy will be conducted. Notably, the TMB detected by whole exome sequencing will be converted to mutations per megabase using linear transformation.[25] R V.4.0.2 will be used for the analyses mentioned above.

## Patient and public involvement

As our study is a protocol of meta-analysis, which based on the previously published literature, the primary patient data will not need to be collected. Existing databases will be used for the purpose of this study. The public or patients will not be involved in the study design, recruitment or data analysis.

## ETHICS AND DISSEMINATION

The data included in this project will be collected from the original studies; therefore, ethical approval and informed consent of patients will not be needed. This systematic review will assess the predict value of TMB in patients with malignant solid tumours. Patients treated with ICIs with high/low levels of TMB will eventually benefit from the knowledge of this study. We will publish the results of this protocol as a complete meta-review paper in an academic journal and scientific conferences.

**Contributors** XX, WG and YL conceived the study and drafted the manuscript. XX and YL registered the protocol systematic review in the PROSPERO database. YL and XY designed the search strategy. PZ, WG and YL formed the data synthesis and analysis plan. XX, XY and YL supervised this study and revised the manuscript.

**Funding** Military Medical Research Project, the General Hospital of Western Theater Command, Chinese People's Liberation Army (2019ZY10, 2021-XZYG-A14); Special Scientific Research Project of Army Health Care (21BJZ39); 2021 Basic Research Cultivation Project of the Central Universities (2682021ZTPY018).

**Competing interests** None declared.

**Patient and public involvement** Patients and/or the public were not involved in the design, or conduct, or reporting, or dissemination plans of this research.

**Patient consent for publication** Not applicable.

**Provenance and peer review** Not commissioned; externally peer reviewed.

**ORCID iD**
Xuemei Xiang http://orcid.org/0000-0001-8803-312X

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
