## [Reviewer comments · BMJ Open]

ARTICLE DETAILS

TITLE (PROVISIONAL)	The Clinical Utility of Tumour Mutational Burden on Efficacy of Immune Checkpoint Inhibitors in Malignant Solid Tumour: Protocol for A Systematic Review and Meta-analysis
AUTHORS	Xiang, Xuemei; Li, Yunming; Yang, Xiaoguang; Guo, Wang; Zhou, Pengfei

VERSION 1 – REVIEW

REVIEWER	Reem Malouf National Perinatal Epidemiology Unit, uffield Department of Population Health
REVIEW RETURNED	19-Mar-2022

GENERAL COMMENTS	This is a PROTOCOL of a systematic review of an important area; The main points you need to address are listed below. 1) The authors used the past tense in almost all of the manuscript, as the work has been completed. It is a protocol of a systematic review and the tense should refer to the future,2) The title is solid tumour and in the abstract is malignant solid tumour?3) This is a protocol of a systematic review however, the authors sometimes referred to this as a review “4) Review primary and secondary objectives are not clear at all. Was the aim to look at ICIs only or the combined therapy with chemo?5) The inclusion criteria is not clear. Please rewrite using the PICO/PICO alternative. Make sure to clarify whether you are planning to look at all solid tumour or only malignant ones.6) The TMB cut off in not clear to distinguish between low/high.7) The analysis method: the authors are planning to use both fixed and random effects, I think they need to choose one and support their choice.8) Clearly list the subgroups the authors are interested in investigating.9) The search strategy: From inception to 31 October 2021, the search should be in the present. Search report is also missing.
---

REVIEWER	Virote Sriuranpong Chulalongkorn University
REVIEW RETURNED	15-May-2022

GENERAL COMMENTS	Several attempts have been done to address the value of TMB as a positive predictive marker for ICI but haven't come to a solid conclusion. There are several possibilities but not all inclusive ie. different platforms and cut off, different ICIs, variabilities of cancer type, combination regimens of ICI. With these reasons, it is difficult to imagine that an update analyses would overturn the prior
---

	conclusions. Authors may have to address several concerns and propose solution to avoid repeating another inconclusive analysis.
--	--

VERSION 1 – AUTHOR RESPONSE

Reviewer: 1

Dr. Reem Malouf, National Perinatal Epidemiology Unit

Comments to the Author:

This is a PROTOCOL of a systematic review of an important area; The main points you need to address are listed below.

1) The authors used the past tense in almost all of the manuscript, as the work has been completed. It is a protocol of a systematic review and the tense should refer to the future,

Yes, we checked and change to the future tense. (Page1-6)

2) The title is solid tumour and in the abstract is malignant solid tumour?

Yes, we checked and used words” malignant solid tumour” in the manuscript title, as we plan to explore malignant solid tumour only. (Page1)

3) This is a protocol of a systematic review however, the authors sometimes referred to this as a review “

Yes, we checked and clearly defined this is systematic review (Page 5, paragraphs 2; Page 6, paragraphs 5)

4) Review primary and secondary objectives are not clear at all. Was the aim to look at ICIs only or the combined therapy with chemo?

Yes, we did not describe our objectives clearly, we aim to look at both ICIs alone and ICIs combined with chemotherapy.

5) The inclusion criteria is not clear. Please rewrite using the PICO/PICO alternative. Make sure to clarify whether you are planning to look at all solid tumour or only malignant ones.

Yes, we did not describe our inclusion criteria clearly, we have been rewritten the PICO/PICO alternative, which is clarified our aim to only malignant solid tumour. (Page 4, paragraphs 1)

6) The TMB cut off in not clear to distinguish between low/high.

Yes, a cutoff of ≥ 10 mutations per megabase (mut/Mb) is chosen to define the “high TMB” patient population.

7) The analysis method: the authors are planning to use both fixed and random effects, I think they need to choose one and support their choice.

Yes, we plan to use fixed-effects model when $I^2 \leq 50\%$ or random-effects model when $I^2 > 50\%$.

8) Clearly list the subgroups the authors are interested in investigating.

Yes, the subgroups as follows: number of patients (High/Low TMB), follow-up period, tumor size, tumor area, stage, cancer type, line of therapy, TMB sequencing method, type of immunotherapy of ICIs alone (PD-L1, PD-1, CTLA-4 et.al) or ICIs combined with chemo.

9) The search strategy: From inception to 31 October 2021, the search should be in the present. Search report is also missing.

Yes, literature search will be changed from inception to 31 May 2022. (Page 2, paragraphs 2). We added search report as below.

Search report (OVID)

Item	Search strategy	Items
1	Immune Checkpoint Inhibitor/ or Immune Checkpoint Inhibition/or Immune Checkpoint Blockers / or Immune Checkpoint Blockade / or PD-1 / or PD-L1/ or CTLA-4/ or nivolumab/ or pembrolizumab / or atezolizumab / or avelumab / or durvalumab/ or tremelimumab / or ipilimumab /	12985
2	mutation/ or mutational/ or burden/ or weight.mp.	1630914
3	1 and 2	700
4	tumor/ or cancer/ or neoplasms.mp.	2969591
5	3 and 4	573

10) Competing interests of Reviewer: None Known

Yes, we added competing interests and there is none competing interests or Reviewer (Page 6, paragraphs 7).

Reviewer: 2

Dr. Virote Sriuranpong, Chulalongkorn University

Comments to the Author:

1) Several attempts have been done to address the value of TMB as a positive predictive marker for ICI but haven't come to a solid conclusion. There are several possibilities but not all inclusive ie. different platforms and cut off, different ICIs, variabilities of cancer type, combination regimens of ICI. With these reasons, it is difficult to imagine that an update analyses would overturn the prior conclusions. Authors may have to address several concerns and propose solution to avoid repeating another inconclusive analysis.

Yes, the value of TMB as a positive predictive marker of ICIs is still controversial. Indeed, there are several factors that influence the clinical efficacy of ICIs ie. different platforms and cut off, different ICIs, variabilities of cancer type, combination regimens of ICI. Therefore, we decide to take a comprehensive subgroup analysis ie. number of patients (High/Low TMB), follow-up period, tumor size, tumor area, stage, cancer type, line of therapy, TMB sequencing method, type of immunotherapy of ICIs alone (PD-L1, PD-1, CTLA-4 et.al) or ICIs combined with chemotherapy. We aim to provide more evidence for evaluating TMB as a predictive biomarker of ICIs.